# The Impact of Script Concordance Testing on Clinical Decision-Making in Paramedic Education

**DOI:** 10.3390/healthcare12020282

**Published:** 2024-01-22

**Authors:** Katarzyna Naylor, Jane Hislop, Kamil Torres, Zakaria A. Mani, Krzysztof Goniewicz

**Affiliations:** 1Independent Unit of Emergency Medical Services and Specialist Emergency, Medical University of Lublin, Chodzki 7, 20-059 Lublin, Poland; 2Clinical Education, Edinburgh Medical School, The University of Edinburgh, Edinburgh EH16 4SB, UK; jane.hislop@ed.ac.uk; 3Department of Didactics and Medical Simulation, Faculty of Medical Sciences, Medical University of Lublin Poland, Chodźki 7, 20-093 Lublin, Poland; kamiltorres@umlub.pl; 4Nursing College, Jazan University, Jazan 45142, Saudi Arabia; zakaria.mani@jazanu.edu.sa; 5Department of Security Studies, Polish Air Force University, 08-521 Dęblin, Poland; k.goniewicz@law.mil.pl

**Keywords:** script concordance test, clinical reasoning, undergraduate paramedic students, formative assessment, educational impact, feasibility, acceptance, summative multiple-choice questions, reliability, validity

## Abstract

This study investigates the effectiveness of the Script Concordance Test (SCT) in enhancing clinical reasoning skills within paramedic education. Focusing on the Medical University of Lublin, we evaluated the SCT’s application across two cohorts of paramedic students, aiming to understand its potential to improve decision-making skills in emergency scenarios. Our approach, informed by Van der Vleuten’s assessment framework, revealed that while the SCT’s correlation with traditional methods like multiple-choice questions (MCQs) was limited, its formative nature significantly contributed to improved performance in summative assessments. These findings suggest that the SCT can be an effective tool in paramedic training, particularly in strengthening cognitive abilities critical for emergency responses. The study underscores the importance of incorporating innovative assessment tools like SCTs in paramedic curricula, not only to enhance clinical reasoning but also to prepare students for effective emergency responses. Our research contributes to the ongoing efforts in refining paramedic education and highlights the need for versatile assessment strategies in preparing future healthcare professionals for diverse clinical challenges.

## 1. Introduction

In the world of medical education, the evaluation of student competence is critical for ensuring the efficacy of teaching and learning processes. While conventional assessment methods such as multiple-choice questions (MCQs) have been a longstanding cornerstone in this domain, these tools are not without their limitations. MCQs, for instance, often focus more on testing factual knowledge and do not necessarily measure higher levels of cognition, such as decision-making or problem-solving abilities [1]. In response to this, innovative assessment tools have been developed to better evaluate the complex cognitive processes that underpin competent clinical practice. One such tool is the Script Concordance Test (SCT) [1,2].

The SCT is a reliable and valid assessment tool that can evaluate how health science students and professionals use their clinical knowledge to make decisions in uncertain contexts [3]. The test has been used extensively in several health science disciplines, including medicine and nursing, to measure and improve clinical reasoning skills, particularly in areas where there is significant uncertainty or where multiple correct answers may exist [4]. The SCT assesses the application of knowledge rather than just its acquisition, focusing more on the ‘how’ rather than the ‘what’ of knowledge use, making it a valuable instrument in evaluating the decision-making capabilities of students [5,6].

In the field of paramedicine, professionals often encounter unpredictable and high-stakes situations where they must quickly and effectively make decisions, often with incomplete or ambiguous information. In this context, the need for efficient clinical decision-making capabilities is paramount, making the SCT an attractive tool for evaluation [7]. However, despite the potential applicability of the SCT to paramedic education, its use and implementation in this field have not been extensively investigated.

Furthermore, the impact of the SCT on students’ learning experiences, particularly in paramedic education, requires further exploration. Studies suggest that the SCT positively influences the learning process, aiding in self-evaluation and identifying knowledge gaps. However, its acceptability among students needs more understanding [5,6]. Moreover, assessing students’ attitudes towards SCTs is crucial. If they do not perceive it as beneficial, its effectiveness might be compromised [7,8,9]. Exploring these attitudes can offer insights into SCT’s optimal integration into curricula and enhancing learning experiences. This exploration is particularly vital given the unique challenges and high-stress nature of paramedic work, where decision-making skills are paramount.

However, as with the adoption of any new tool or approach, it is necessary to consider the specifics of the setting in which it will be implemented. For example, in paramedic education, courses are often designed to be highly practical, placing a significant emphasis on real-world, hands-on experiences. As such, any new assessment methods should be carefully calibrated to align with these practical elements and ensure that they are testing the right capabilities. While the SCT has shown promise in other fields of healthcare education, its implementation within the unique context of paramedic education requires careful consideration and thorough investigation.

After this consideration, the present study seeks to examine the potential of the SCT as an assessment tool within paramedic education. It investigates the implementation of the SCT in a qualified first aid course and its correlation with traditional MCQ-based assessments. Furthermore, this study takes into account students’ experiences with the SCT, thereby contributing to a better understanding of its potential role and impact within the framework of paramedic education.

## 2. Materials and Methods

### 2.1. Design and Setting

The study employed a prospective cohort design, focusing on pre- and post-intervention assessments. Specifically, the SCT was administered to two cohorts of paramedic students both before and after their participation in a specialized first aid course. This approach allowed for a focused examination of the SCT’s impact over a defined period. The study was conducted from June 2021 to April 2022 at the Faculty of Medicine, Medical University of Lublin, in Poland.

The particular first aid course that the students underwent was a qualified one, geared towards honing their skills and building upon the knowledge they acquired from a preliminary first aid course taken earlier in their academic journey. A significant emphasis of this course was on attending to trauma patients and adhering to international trauma life support guidelines in a prehospital setting. It was structured and executed in compliance with the learning objectives delineated in the Field of Qualified First Aid National Bill [10] and the International Trauma Life Support recommendations [11].

The physical setting for this investigation, the Medical University of Lublin, offered a conducive environment for learning and testing. The students, immersed in a setting echoing real-world dynamics, had a chance to connect theory with practice more effectively. The confluence of a strategic study design and a practical setting played an instrumental role in enabling an in-depth exploration of the learning outcomes related to the qualified first aid course.

### 2.2. Recruitment and Sample

The study utilized a convenience sampling strategy, recruiting students from the Medical University of Lublin’s (MUL) first-year (Cohort I, CI, *n* = 31) and second-year (Cohort II, CII, *n* = 38) paramedic programs, yielding a combined total of 69 participants. Further, summative MCQ examination results from a third group (Cohort III, CIII, *n* = 30), consisting of students who had finished the qualified first aid course during the prior academic year, were included to supplement the data (Figure 1). This group functioned as a control for comparison. The strategy aimed to create a participant pool that was homogenous in their educational background and experiences, thereby facilitating more robust and valid conclusions.

### 2.3. Inclusion and Exclusion Criteria

For the recruitment process of our study, we utilized convenience sampling, a common approach in academic research, particularly within the constraints and practicalities of medical education settings. We invited all first-year (Cohort I, CI, *n* = 31) and second-year (Cohort II, CII, *n* = 38) paramedic students at the Medical University of Lublin (MUL) to participate. These students constituted the primary cohorts of our study, totaling 69 participants.

In addition, our analysis incorporated data from a third cohort (Cohort III, CIII, *n* = 30)—a group of students who had completed their qualified first aid training in the previous academic year. This inclusion offered a basis for comparative analysis, enriching the scope of our findings by providing an additional reference point.

The chosen sampling approach facilitated the collection of data from a readily accessible participant pool within our institutional setting. This allowed for an efficient gathering of a diverse and contextually rich dataset, enabling a multifaceted examination and comprehensive interpretation of our research findings.

This revision presents the convenience sampling method in a neutral and factual manner, acknowledging its common use in academic research, especially within medical education contexts, without overstating its benefits. It also clearly describes the composition of the cohorts and the rationale behind including a third cohort for comparative analysis.

#### 2.3.1. Inclusion Criteria

The study focused on undergraduate paramedic students at MUL who were in their first or second year and were either due to undertake or had already completed the qualified first aid course. This ensured that participants were at a similar level of initial knowledge, making the results more comparable. Additionally, all participants were required to have completed an introductory first aid training course prior to the qualified first aid course, ensuring a foundational understanding of first aid principles.

#### 2.3.2. Exclusion Criteria

Participants were excluded if they had previously received diploma-certified training in qualified first aid to avoid the confounding effects of advanced prior knowledge. Additionally, individuals with other certifications related to paramedic qualifications, such as a college paramedic diploma, were excluded. This was to ensure that additional qualifications did not influence their performance in the SCT or their progression during the first aid course.

### 2.4. Ethical Considerations

The study obtained ethical approval from the Bioethics Committee at the Medical University of Lublin (approval number: KE-0254/154/2020) and was conducted in line with the ethical principles articulated in the Recommendations from the Association of Internet Researchers.

### 2.5. Script Concordance Test

The SCT was used as a primary tool for evaluating the students’ clinical decision-making abilities. This written questionnaire, derived from previous research studies [4], was designed to align with the content and learning objectives of the qualified first aid course. The SCT allows for the assessment of clinical judgment skills in uncertain clinical scenarios, which are crucial for future paramedics dealing with trauma patients.

The SCT featured six distinct scenarios, each containing three questions focusing on further management. The essence of the SCT lies in its design: each scenario is subsequently supplemented by a new piece of information, potentially altering the initial management plan [12]. The use of a 3-point Likert scale, as recommended by Fournier et al. [12], allowed us to gauge the students’ responses based on their degree of agreement or disagreement with the given statements (Appendix A).

This test was primarily aimed at first-year and second-year students who are still in their early stages of learning and have yet to be exposed to more advanced medical scenarios. The purpose was to introduce them to real-world situations where clinical decisions often need to be made in uncertain circumstances, thereby fostering their capacity to think critically and make well-informed decisions when faced with practical medical emergencies. This approach contributes significantly to the effective learning and understanding of the field’s practical aspects, which directly aligns with the pedagogical goals of the qualified first aid course at MUL.

### 2.6. Data Collection

In our study, the data collection process involved two phases of SCT administration for students in Cohorts I and II. Initially, unrestricted online access to the SCT was provided for a five-day period prior to their first qualified first-aid lecture. The purpose was to establish a baseline for students’ knowledge and clinical reasoning in first aid.

Following the course, these students had a second opportunity to take the SCT, aimed at assessing the impact of the course on their understanding and application of first aid principles.

The SCT completion process was intentionally designed to be student-friendly. We chose to allow students to complete the test at their convenience in a comfortable environment, diverging from the high-pressure, unpredictable scenarios typically associated with paramedic work. This approach was adopted to ensure the accuracy of responses and minimize stress, which can significantly influence cognitive performance. By providing a relaxed setting, we aimed to obtain a clear measure of their clinical reasoning skills without the confounding effects of stress or time pressure. This methodology, though seemingly contrary to the realities of paramedic work, was instrumental in isolating and understanding the pure cognitive and decision-making abilities of the students.

Participation in the SCT was voluntary, with students informed that their consent for participation would also include the use of their data in the study.

To evaluate student acceptance of the SCT, we conducted an anonymous survey using The Utrecht Seminar Evaluation (USEME) questionnaire post-course. This survey was administered online via Google Surveys, a recognized method in healthcare research for collecting respondent data [13]. The use of the USEME questionnaire in our research received approval from Spruijt et al. [14] (See Appendix B for details).

### 2.7. Data Analysis

Our data analysis process commenced with organizing the collected data using Microsoft Excel (2020) and creating a comprehensive database encompassing SCT, MCQ results, and USEME questionnaire responses.

For the statistical analysis, we employed STATISTICA 10 (StatSoft, Kraków, Poland) due to its extensive capabilities. Initially, categorical variables were described using numbers and percentages. For quantitative variables, we used measures such as mean value (M), standard deviation (SD), median (Me), interquartile range (IQR), minimum (Min), and maximum (Max) to describe central tendency and dispersion. The inclusion of the IQR was particularly crucial for data not following a normal distribution.

To assess the normality of the data distribution, we employed the Shapiro–Wilk test, with a significance level set at *p* < 0.05. For examining the concurrent validity between MCQ and SCT scores, the Bland–Altman method, including scatter plots, was utilized to assess the agreement between these assessment methodologies.

Additionally, we explored the relationship between MCQ and SCT scores using the nonparametric Spearman rank-order correlation coefficient, facilitating comparisons with previous studies. The significance of these comparisons was determined using *p*-values less than 0.05.

Importantly, for evaluating the internal consistency of the SCT, we used Cronbach’s alpha coefficient. This metric was critical to assess the reliability and homogeneity of the test items within the SCT, ensuring the assessment’s robustness and appropriateness for our study objectives.

Furthermore, reliability analysis included the use of intraclass correlation coefficients (ICCs) and the coefficient of variance, which were instrumental in evaluating the educational impact of the SCT.

Finally, the SCT’s ‘acceptability’ was assessed through data obtained from the USEME questionnaire. This thorough and methodical approach to data analysis was meticulously designed to maximize the extraction of insights and information from the collected data.

## 3. Results

Among the 69 paramedic students initially considered, complete SCT data, MCQ exam results, and post-training evaluation questionnaire responses were gathered for 55 students, 24 from CI and 31 from CII. This corresponds to a return rate of 80%, forming the backbone of our final analysis. Additionally, we incorporated the MCQ results for 30 students from CIII into the analysis. Figure 2 outlines the recruitment process and the final cohort numbers included in the analysis.

### 3.1. Normality of the Data

The data were analyzed using nonparametric tests as indicated by the Shapiro–Wilk test, where the MCQ results and the SCT results significantly deviated from a normal distribution (*p* < 0.0001). Consequently, the median and Interquartile Range (IQR) are reported [15].

### 3.2. Concurrent Validity

A Spearman rank-order correlation coefficient was employed to examine the relationship between SCT and MCQ scores for CI and CII. The analysis indicated no significant correlation between the SCT results and MCQ results (rs = 0.18; *p* = 0.2).

To investigate the concurrent validity further, we conducted Bland–Altman plots comparing SCT and MCQ examination results (Figure 3). The analysis indicated a mean difference of about 14% between the results of the two methods, with MCQ results being, on average, 14% higher than those of the SCT. However, the wide limits of agreement (LOA ± 1.96 SD: 41.7% to −13.4%) and the fact that zero is within these limits suggest there is no meaningful difference between the two measures. The LOA was also visibly dispersed, indicating a weak concurrent agreement between the SCT and MCQ scores (Table 1).

### 3.3. Reliability Analysis

Intraclass Correlation Coefficients (ICCs) were calculated to examine the agreement between the scores for CI and CII, along with providing the 95% confidence intervals [CI]. The SCT results before the qualified first aid course showed an ICC of 0.90 [CI: 0.90 to 0.99] for single measures and 0.95 [CI: 0.90 to 0.99] for average measures, demonstrating a high level of agreement. In contrast, the post-course SCT results showed ‘moderate’ relative reliability (ICC: 0.45 [CI: −0.25 to 0.77] for single measures and 0.55 [CI: −0.25 to 0.77] for average measures).

The variability coefficient in the MCQs and the SCT was comparable (13.7% vs. 13.9%). Details are presented in Table 2.

### 3.4. Internal Consistency

We utilized Cronbach’s α coefficient to evaluate SCT’s internal consistency. The resulting Cronbach’s α coefficient was 0.67, suggesting a satisfactory level of internal consistency for the SCT [16,17,18].

### 3.5. Educational Impact

To assess the educational impact of the SCT, the MCQ results for CI and CII (those who had undergone an SCT as part of their training) were compared with the MCQ results from CIII (those who received their training in the previous academic year without the SCT). The findings revealed a statistically significant difference between the median percentage results of the two groups (i.e., CI/CII and CIII) (*p* < 0.001). In particular, the mean MCQ scores obtained by the CI and CII (SCT group) were significantly higher than those of the CIII group (control group). Figure 4 illustrates a box plot of the differences in MCQ results for CI/CII as compared to the MCQ results for CIII.

The educational impact was further probed by analyzing two attempts of the SCT for both cohorts: CI and CII. A nonparametric Wilcoxon test was utilized to compare these two assessment points.

The comparison results in CI before and after the certified first aid course showed no statistically significant difference in the total points scored at these two assessment points (T = 148.000; *p* > 0.5). These results are displayed in Table 3.

In contrast, the comparison in CII before and after the certified first aid course demonstrated a statistically significant difference in the total points scored between these two assessment points (z = 0.7; *p* < 0.5), implying an increase in scores after the first aid course. However, due to the small sample size, this difference was not as discernible. These results are presented in Table 4.

The Wilcoxon test results suggested no statistically significant difference in the SCT scores between CI and CII (z = 0.3 *p* > 0.5) before the course. Similarly, no significant difference was found in the post-course SCT results between CI and CII (z = 0.8; *p* > 0.05).

When grouping both cohorts (CI and CII) and comparing results before and after the certified first aid course, the findings revealed no statistically significant difference in the total points scored at these two assessment points (z = 0,3; *p* > 0.5). These results are exhibited in Table 5.

### 3.6. Acceptability

Data from the USEME questionnaire was collected from all participants in CI and CII at the conclusion of the certified first aid course [12]. The questionnaire consisted of 17 statements related to the SCT, and participants rated their responses on a scale from 0 (strongly disagree) to 5 (strongly agree).

Feedback regarding the certified first aid course and its content was generally favorable across all questions, with mean scores exceeding four for all queries. The participants reported finding the online SCT to be a valuable part of the course and expressed a desire to continue using such assessments in the future.

## 4. Discussion

Clinical reasoning forms the backbone of healthcare, enabling professionals to make timely and beneficial decisions [19]. However, many factors influence clinical reasoning, making it a multifaceted, complex, and often elusive process [7]. Recognizing these challenges, our study aims to contribute to the ongoing exploration of clinical reasoning. Through a prospective cohort study involving a specific group of 55 paramedic students, we explored the utility of the SCT in stimulating and evaluating clinical reasoning. While our findings provide valuable insights, they should be viewed as preliminary, given the limited sample size and the specific context of a single faculty at a particular university. As such, this study represents an exploratory step in understanding the application of SCTs in paramedic education.

Our results indicate that the SCT is a promising tool in paramedic education, with potential reliability and effectiveness. While the direct impact of the SCT on educational outcomes, as measured by pre-post-test comparisons, was not distinctly evident, participants did report meaningful engagement with the content through the SCT. This engagement is reflected in their substantial alignment with the course material when interacting with the SCT. Notably, there was a variance in how first-year and second-year students perceived the SCT’s utility in understanding course material, with second-year students showing a more favorable response. This difference may be attributed to the second-year students’ increased familiarity with the SCT format, suggesting a potential benefit in introducing the SCT earlier in the curriculum to enhance its effectiveness and student comfort.

Our study provides valuable insights into SCT’s validity, specifically its concurrent validity. We noted that the median MCQ scores of the participants surpassed those of the SCT. This gap could stem from the unfamiliarity of the students with the SCT, as it presents a significant departure from traditional assessment methods. The SCT demands a higher level of cognitive engagement from students, as it prompts them to make decisions along a Likert scale, contrasting the binary choices presented by MCQs. This novel method, while challenging, helps to evaluate not just factual recall but the critical thinking process involved in decision-making [20,21].

In our analysis, notable differences were observed between Cohort I (first-year students) and Cohort II (second-year students) in their response to and performance on the SCT. These variations may be attributed to several factors, including the differing levels of exposure and experience with the clinical environment and decision-making processes. Second-year students, having had more time to acclimatize to the academic and practical aspects of paramedic training, might be better equipped to handle the complexities of the SCT. This disparity underscores the importance of progressive and scaffolded learning approaches in paramedic education, where students gradually build their competencies over time. Additionally, these findings suggest the need for early and continuous exposure to diverse assessment tools, like the SCT, throughout the paramedic curriculum. This approach could facilitate a more uniform development of clinical reasoning skills, bridging the gap observed between different year cohorts.

In this study, we addressed two distinct yet interconnected aspects of the SCT in paramedic education. Firstly, we evaluated the SCT’s capacity to meaningfully assess students’ knowledge and clinical reasoning. Our findings indicate that the SCT offers a nuanced approach to understanding student comprehension and decision-making, particularly in complex and uncertain scenarios characteristic of paramedic practice. Secondly, we explored the SCT’s influence on the learning trajectory of students. The data suggest that engagement with SCTs potentially enhances students’ propensity to learn and adapt, fostering a deeper level of cognitive engagement and critical thinking. This dual focus—on both assessment and learning enhancement—is critical in understanding the broader educational impact of the SCT. By differentiating these roles, we provide a clearer picture of SCT’s multifaceted contribution to paramedic education, offering insights into how it can shape both evaluation methods and learning processes.

In considering the evolution of clinical reasoning assessment, it is noteworthy that the concept has undergone significant transformation over the years [22]. Historically, clinical reasoning was predominantly assessed through oral examinations and practical demonstrations, which gradually evolved into more structured formats like MCQs and OSCEs. The introduction of the SCT marked a further advancement in this evolution, offering a more nuanced approach to evaluating clinical reasoning skills [23]. More recently, the integration of digital technology in assessment methods, such as the use of virtual patient simulations, has opened new avenues for evaluating and enhancing clinical reasoning skills in a dynamic healthcare landscape [24]. This historical progression underscores the importance of continuous innovation in educational methodologies to keep pace with the evolving demands of medical training.

Research by Tan et al. and Delavari et al. supports our contention, demonstrating the SCT is a powerful complement to traditional assessment methods, enriching fields like neurology and midwifery among medical students [25,26]. The SCT was also employed effectively in conjunction with flipped classrooms and gamification to stimulate self-directed learning among anatomy students [25]. Based on our findings and a review of the relevant literature, we strongly advocate for SCT’s integration alongside traditional assessment methods. This is especially pertinent in areas that require distinct critical thinking and reasoning skills, in which the SCT could provide valuable learning support.

The correlation between SCTs and traditional assessment methods like MCQ is another area our research illuminates. Similar to Goos et al.’s findings in visceral surgery among medical students, we found no significant correlation [17]. This aligns with Duggan and Charlin’s conclusions, which revealed a weak to moderate correlation between MCQ and OSCE in multidisciplinary assessments [18]. Beyond paramedic education, the SCT approach has potential applicability in various other healthcare disciplines. For instance, its use has been explored in nursing education to enhance clinical judgment and decision-making skills, as evidenced by the work of Mukhalalati [19]. Similarly, in clinical education, the SCT has been proposed as a means to assess therapeutic decision-making abilities, as discussed by Ross et al. [7]. Such observations further stress the importance of early exposure to the SCT in the learning process, enabling students to familiarize themselves with its distinct methodologies. In the context of medical and allied health education, early integration of the SCT could lead to more effective and holistic learning experiences. This strategy would allow for more meaningful SCT deployment in the final years of medical studies, potentially contributing to a more adaptive and competent healthcare workforce capable of meeting diverse clinical challenges.

Lubarsky et al. and Humbert et al. substantiate the content validity of the SCT through its ability to differentiate between various levels of expertise and the extent to which it represents targeted medical areas [27,28]. Mirroring this approach, we sought to confirm the content validity of the SCT in the current study. Regular discussions among educators during the SCT item development stage ensured a comprehensive representation of the qualified first aid course material, thereby reinforcing the content validity of the SCT.

A compelling measure of SCT’s success is its reliability. Our SCT showed satisfactory reliability levels based on ICC results, coefficient of variance, and Cronbach’s alpha coefficient for internal consistency (Cronbach’s α = 0.67). Other researchers, including Wan et al., Ang et al., Goos et al., Kaur et al., Lubarsky et al., Humbert et al., and Mathieu et al., have found comparable reliability figures [17,27,28,29,30,31,32]. However, Tavakol and Dennik caution against relying solely on Cronbach’s alpha due to its vulnerability to sample size and test length effects [33]. Consequently, our reliability analysis incorporated other statistical measures like ICC and CV, which provided more robust validity evidence.

Our research expands the application of the SCT beyond medical doctors and nursing professionals to paramedic students. Consequently, our findings have broader implications for education in healthcare. As healthcare becomes increasingly complex, developing and assessing clinical reasoning skills becomes critical. The SCT could serve as an effective method to achieve this, complementing traditional assessment tools and providing a more nuanced understanding of a student’s learning progress [34,35,36,37,38,39,40,41,42,43].

We propose that future research should explore the relationship between SCT performance and actual professional performance among paramedics. This line of inquiry could potentially unearth valuable findings for both educational and professional domains in paramedicine. Additionally, future studies could involve the development of a definitive benchmark for SCTs, which could serve as a comparison point in understanding SCT scores.

This study sheds new light on the validity, reliability, and educational impact of SCTs in paramedic education. The SCT has the potential to play a crucial role in the pedagogical shift from rote learning to a more skill-based, decision-oriented educational paradigm. As we strive for this transformation, it is essential that assessment methods evolve to mirror these changes, ensuring that they effectively gauge the spectrum of cognitive skills necessary in modern healthcare.

## 5. Limitations

This study, conducted in a single academic institution, faces limitations in generalizability. Our findings, derived from a specific cohort of 55 paramedic students, offer initial insights but may not be broadly applicable to the entire paramedic student population. Further research involving diverse institutions is necessary to validate and extend our understanding of SCT’s effectiveness in paramedic education.

The prospective cohort design of our study reveals certain relational trends but does not establish definitive cause-and-effect relationships. Therefore, our observations should be considered as preliminary indications, meriting additional investigation.

We also recognize the possibility of overlooked confounding variables inherent in many research designs. This necessitates a cautious interpretation of our results and suggests the use of more comprehensive methodologies in future studies for better identification and control of potential confounders.

Variations in the first aid course curriculum among the cohorts may have influenced the outcomes, indicating the need for standardized curricular content in future research for more consistent results.

The initial unfamiliarity of participants and panel members with SCTs highlights the need for preparatory phases, such as pilot testing, to facilitate smoother integration and refinement of this assessment method in the curriculum.

An early introduction of a simplified SCT in the paramedic education program could enhance its effectiveness, allowing students to acclimate to this approach from the outset of their training.

In conclusion, while our study offers initial insights into the potential of SCTs in paramedic education, it predominantly serves as a foundation for future, more extensive research to substantiate and expand upon these preliminary findings.

## 6. Conclusions

Our study adds to the evidence supporting the SCT as an innovative tool in paramedic education. SCT’s potential to enhance students’ learning processes, particularly in clinical reasoning, has been demonstrated, though our findings regarding its impact on MCQ performance suggest the need for further investigation due to potential confounding factors in cohort comparisons.

The study highlights the SCT’s acceptance among students and its effectiveness in assessing not just knowledge but critical thinking and decision-making skills. While the SCT’s implementation in a Polish university setting shows feasibility, its development demands significant resources, including the creation of authentic scenarios and expert panel involvement.

In summary, our findings point to the promise of SCTs in improving paramedic and potentially other healthcare education programs. The integration of SCTs into healthcare education could mark a significant step in pedagogical evolution, equipping students with essential skills for modern healthcare challenges.

## Figures and Tables

**Figure 1 healthcare-12-00282-f001:**
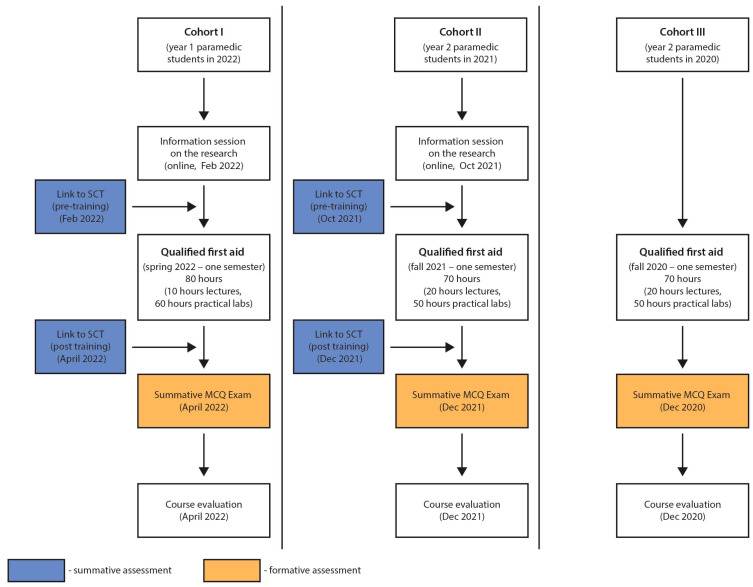
A flow-chart demonstrating the stages and cohorts of the research.

**Figure 2 healthcare-12-00282-f002:**
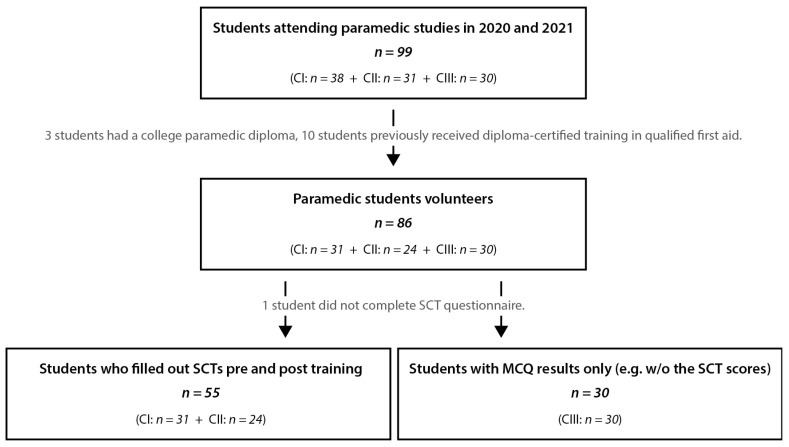
Outline of the recruitment process and the participants included in the research. CI—Cohort 1, year 1 paramedic students from 2021 cohort; CII—Cohort 2, year 2 paramedic student from 2020 cohort; CIII—Cohort 3, year 2 paramedic student from 2019 cohort. SCT—Script Concordance Test.

**Figure 3 healthcare-12-00282-f003:**
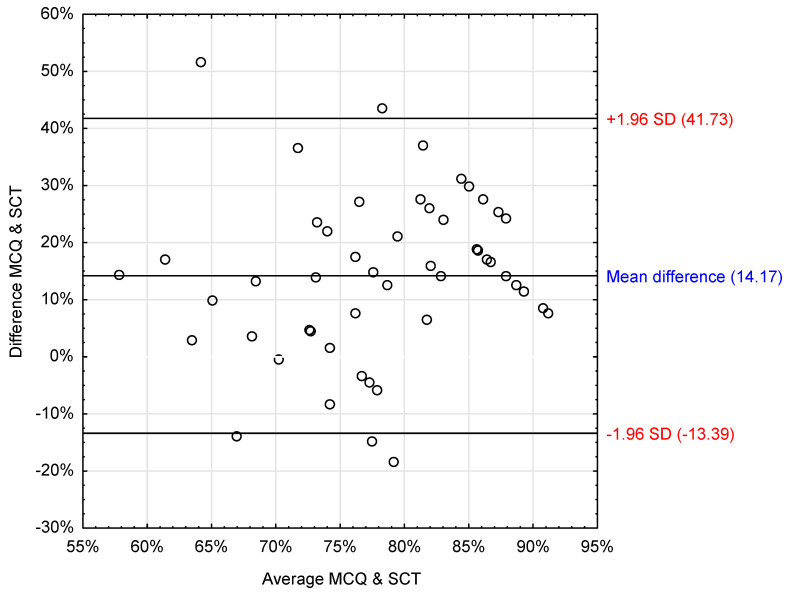
Bland and Altman scatter plot presenting the difference between STC and MCQ results. MCQ: Multiple-Choice Questions, SCT: Script Concordance Test.

**Figure 4 healthcare-12-00282-f004:**
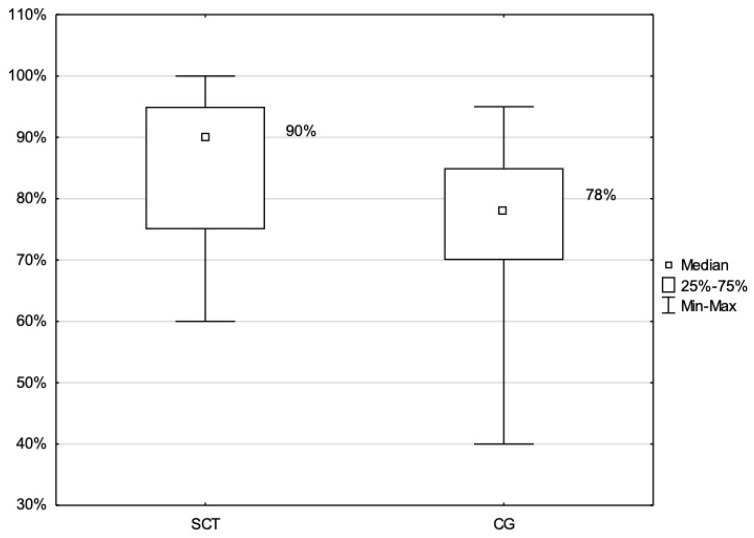
CQ Score Distribution: Comparison between Cohorts with SCT Exposure (I and II) and Non-Exposed Control Group (III). CG: Control group (CIII), MCQ: Multiple-Choice Questions, SCT: Script Concordance Test Group (CI and CII).

**Table 1 healthcare-12-00282-t001:** Mean difference and LOA between MCQ and SCT results.

*n* = 55	Mean Difference %	Upper and Lower LOA 1.96 ± SD %
Exam results [%]	14.2	41.7–13.4

MCQ: Multiple-Choice Questions, SCT: Script Concordance Test, LOA: Limits of agreement.

**Table 2 healthcare-12-00282-t002:** The Coefficient of variance statistics in case of MCQ and the SCT results.

	*n*	M	Min	Max	IQR	CV
SCT	55	65%	6.9	15.9	2.2	13.9%
MCQ	55	85%	12	20	3.0	13.7%

CV—coefficient of variance; M—median; IQR—Interquartile Range.

**Table 3 healthcare-12-00282-t003:** The comparison between the SCT results post and prior to the qualified first aid course in Cohort I.

*n* *	Scores	Me	Min–Max	IQR
24	Pre	12.8	8.8–15.3	11.7–14.1
	Post	12.9	9.1–15.9	11.2–14.1

*n* *—number of paired results; Me—median; IQR—interquartile range.

**Table 4 healthcare-12-00282-t004:** The comparison between the SCT results pre- and post- the qualified first aid course in Cohort II.

*n* *	Scores	Me	Min–Max	IQR
30	Pre	12.6	8.9–15.3	11.9–14.1
	Post	13.2	6.9–15.3	12.1–14.1

*n* *—number of paired results; Me—median; IQR—interquartile range.

**Table 5 healthcare-12-00282-t005:** The comparison between the SCT results post and prior to the qualified first aid course in both cohorts.

*n* *	Scores	Median	Min–Max	IQR
55	Pre	12.6	8.9–15.3	11.80–14.10
	Post	12.89	6.90–15.90	11.90–14.10

*n* *—number of paired results; Me—median; IQR—interquartile range.

## Data Availability

The datasets used and/or analyzed during the current study are available from the corresponding author on reasonable request.

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
