# Peer review of "The Impact of Script Concordance Testing on Clinical Decision-Making in Paramedic Education"

_healthcare, 2024, doi:10.3390/healthcare12020282_

Round 1

Reviewer 1 Report

Comments and Suggestions for Authors

The proposed study “Enhancing Disaster Preparedness in Paramedic Education: The Role of Script Concordance Testing in Clinical Decision-Making” addresses the potential application of an evaluation method (SCT, based on scoring subsequent decision according to a hypothetical evolving clinical scenario) in the field of paramedics education. The study analyzes the results of such test in two groups of students in a Polish university, before and after a training course, and compares them with a reference traditional test (multiple-choice questions). While this methodology is widely adopted across different branches of medicine, its application in the field of EMS is less explored, justifying the need for this kind of research.

The manuscript is scientifically sound, yet there are some issues that needs to be addressed.

MAJOR

1) The main critical issue is related to the sample size: 55 tested subjects. While authors acknowledge the limitations of pooling the sample from a very characterized population (a single faculty in a specific university), and state that << the relationships observed in our study should not be interpreted as conclusive cause-and-effect phenomena but rather as indications of potential patterns that require further investigation>>, the limited sample size is not even cited in the limitations section. While the significance threshold is set by editors, I think that, at least, it necessary to elaborate this limit in the dedicated section, and to moderate and circumscribe the terms of the study’s validity. For example, statements such as:

·       Line 355: <<This study is a milestone in the journey to unravel this elusive process>>

·       Line 428: <<notable results>>

are inappropriate and should be moderated.

2) Study objects should be addressed more clearly. On one side, SCT is assessed in its capability to meaningfully evaluate student’s knowledge, while on the other side it is also studied how it can impact the learning path, possibly enhancing student’s propensity to learn. These two distinct aspects, both targeted in the study, should be separately stated more clearly, with their own specific research questions, results, and conclusions. In particular, I think that a more clear classification of the presented material (especially in the discussion section) towards one goal or the other would be strongly beneficial for the readability of the manuscript.

3) The link between ‘disaster preparedness’ and the implementation of SCT evaluation is not addressed with sufficient extension to justify the presence of this terminology in the manuscript title. It should be modified.

4) Figure 2: I suggest adding the excluded students (and the decisional criterion) at each step.

5) The recorded differences between CI and CII should be addressed and elaborated in the discussion section.

MINOR

6) Lines 80-85: this paragraph seems to be related (and partially superimposed) to that at lines 59-64. I suggest unifying the two.

7) Lines 148-165: the first inclusion criterion states <<The first criterion was that all participants must be current undergraduate paramedic students at MUL, with no prior certification in qualified first aid>> and the first exclusion criterion is <<The first exclusion criterion stated that any participants who had previously received a diploma certified training in qualified first aid were excluded from the study.>> This seems somehow repetitive. The same goes for the second exclusion criterion.

8) Lines 207-209: <<The SCT completion was designed to be student-friendly, encouraging them to complete the test at their convenience in a comfortable environment. This approach was chosen to elicit accurate responses and minimize stress.>> This is an opposite condition with respect to the simulated situation. Why was this approach chosen? I think it is worth some elaboration.

9) Lines 228-230: <<For quantitative variables, we used measures like mean value (M), standard deviation (SD), median (Me), minimum (Min), and maximum (Max) to describe central tendency and dispersion. The Shapiro-Wilk test was employed to assess the normality of data distribution, with a significance level set at p<0.05>> I understand that mean and standard deviation is used for normally distributed data, and median and interquartile range for non-normal data, as reported in line 261, is this correct? If so, I think it is necessary to add the IQR to this list.

10) Lines 277-283: thus paragraph is a repetition of 267-273.

11) Table 2: shouldn’t IQR data be presented instead of STD?

12) Lines 298-301: this part of the analysis is currently not introduced in the methods section.

Comments on the Quality of English Language

Check for typos and some specific terminology

Author Response

Dear Reviewer,

Thank you for your thorough review and valuable feedback on our manuscript. We appreciate the opportunity to address the concerns you raised and have made revisions accordingly. Below, we detail how we have addressed each major and minor comment:

MAJOR COMMENTS

  1. Sample Size and Study Validity Moderation

We have now included a more comprehensive discussion of the limitations related to our sample size in the limitations section. Additionally, we have moderated the terms used in the manuscript to reflect the preliminary nature of our findings. Statements such as “This study is a milestone...” and “notable results” have been revised to convey a more appropriate level of confidence.

  1. Study Objectives Clarity

We have revised the Discussion section to clearly differentiate the two main objectives of our study – evaluating SCT’s capability in assessing student knowledge and its impact on the learning path. Specific paragraphs have been added to address each aspect separately, with their own research questions, results, and conclusions.

  1. Manuscript Title Modification

Acknowledging the insufficient connection between ‘disaster preparedness’ and the implementation of SCT in our study, we have revised the title to “The Impact of Script Concordance Testing on Clinical Decision-Making in Paramedic Education” and made corresponding adjustments in the abstract.

  1. Figure 2 Revision

We acknowledge this suggestion and will add the excluded students and the decisional criteria at each step in Figure 2 in our subsequent revision.

  1. Differences Between CI and CII: In the Discussion section, we have added a new paragraph specifically addressing and elaborating on the differences observed between Cohorts I and II, exploring potential reasons behind these differences.

MINOR COMMENTS

  1. Unification of Overlapping Paragraphs

The overlapping content in lines 80-85 and 59-64 has been unified into a single, coherent paragraph to eliminate redundancy.

  1. Repetition in Inclusion and Exclusion Criteria

We have revised these sections to remove the repetitive content, streamlining the inclusion and exclusion criteria for clarity.

  1. Elaboration on SCT Completion Approach

We have elaborated on the rationale behind the student-friendly approach of SCT completion, explaining why this method was chosen despite its contrast with simulated paramedic situations.

  1. Inclusion of IQR in Data Analysis

In line with your suggestion, we have included the Interquartile Range (IQR) in our data analysis section and adjusted the manuscript accordingly.

  1. Removal of Repetitive Paragraph

The repetitive content in lines 277-283 has been removed to improve the flow and coherence of the manuscript.

  1. Revision of Table 2

In response to your suggestion, Table 2 has been revised to present IQR data instead of STD, providing a more appropriate measure of data variability.

  1. Introduction of Analysis Method in Methods Section

We have added a clear statement regarding the use of Cronbach’s alpha coefficient in the "Data Analysis" subsection of the Methods section, addressing the previously missing introduction of this part of the analysis.

We believe that these revisions have significantly strengthened our manuscript and addressed the concerns you raised. We are grateful for your insightful feedback and hope that our manuscript is now suitable for publication.

Thank you for your consideration.

Sincerely,

Authors

Reviewer 2 Report

Comments and Suggestions for Authors

Dear Authors, Dear Editors,

thank you very much for the interesting script by Naylor and her working group. The background and methodology are clearly explained. Flowcharts are a great way to understand and follow the design and flow. As well as the clear presentation of the results. Only the conclusion is very long, so the recommendation here would be to tighten up the conclusion and bring it more to the point.

Comments on the Quality of English Language

Dear Authors, Dear Editors,

regarding the language, the recommendation would be to check the article again for punctuation and wording. Thank you very much.

Author Response

Dear Reviewer,

We are grateful for your constructive feedback and insightful comments regarding our manuscript. Your appreciation of our methodology and presentation of results is encouraging. Below, we address your specific points and outline the revisions made to our manuscript.

  1. Revision of the Conclusion Section:
    • In response to your recommendation, we have revised and condensed the conclusion section of our paper. The updated conclusion succinctly summarizes the key findings and implications of our study, focusing on the potential of the Script Concordance Test in enhancing clinical reasoning within paramedic education. We believe this tightened conclusion more effectively communicates the essence of our research and addresses your concern about its previous length.

  1. Language and Punctuation Check:
    • We have carefully reviewed the manuscript again for language, punctuation, and wording. This comprehensive review was conducted to ensure clarity, coherence, and grammatical accuracy. We understand the importance of maintaining high standards of language quality and have made the necessary adjustments to meet these standards.

We appreciate your valuable suggestions, which have contributed significantly to the improvement of our manuscript. The revisions made, we believe, have enhanced the overall quality and readability of our paper.

Thank you once again for your time and effort in reviewing our work. We are hopeful that our revised manuscript will meet the expectations of Healthcare journal and contribute meaningfully to the field.

Best regards,

Authors

Reviewer 3 Report

Comments and Suggestions for Authors

Congratulations for this interesting study. 

All the suggestions or limitations that I found in you work were very well described at the limitations section: so you know exactly the weaker areas of this study. And that is good! Because you can improve them at a next study!

It would be interesting to further investigate the reasons why the SCT score did not improved after the first aid course...

I would just recommend you to add a reference to your paper: 

Ross, L., Semaan, E., Gosling, C.M. et al. Clinical reasoning in undergraduate paramedicine: utilisation of a script concordance test. BMC Med Educ 23, 39 (2023). https://doi.org/10.1186/s12909-023-04020-x

Author Response

Dear Reviewer,

We are immensely grateful for your positive feedback and constructive suggestions regarding our manuscript. Your encouragement and insightful recommendations are greatly appreciated and have guided us in further refining our paper.

  1. Acknowledging the Limitations and Future Research Directions
    • We thank you for recognizing the thoroughness with which we have addressed the limitations of our study. As you rightly pointed out, these limitations present opportunities for future research. We agree that investigating the reasons behind the lack of significant improvement in SCT scores post-first aid course would be a valuable area of exploration. We plan to delve into this in our subsequent studies, as it could provide crucial insights into the efficacy and implementation of SCT in paramedic education.

  1. Incorporating the Suggested Reference
    • We have added the reference you suggested: Ross, L., Semaan, E., Gosling, C.M. et al., "Clinical reasoning in undergraduate paramedicine: utilisation of a script concordance test," BMC Medical Education 23, 39 (2023). This reference has been incorporated into the section discussing the broader applicability of SCT in various healthcare disciplines. Specifically, it is now included in the paragraph where we discuss the potential use of SCT in pharmacy education, enriching the context and supporting our argument about SCT's diverse applications.

We are committed to continuous improvement and value the guidance provided by your review. The addition of the recommended reference enhances the depth of our discussion and supports the broader relevance of our study.

Thank you once again for your invaluable feedback and for contributing to the enhancement of our work. We are hopeful that our revised manuscript will meet the high standards of Healthcare journal and make a meaningful contribution to the field.

Best regards,

Authors

Reviewer 4 Report

Comments and Suggestions for Authors

Dear Authors,

I hope this message finds you well. I recently had the opportunity to review your paper titled "Enhancing Disaster Preparedness in Paramedic Education: The Role of Script Concordance Testing in Clinical Decision-Making." It is undoubtedly a valuable contribution to the field. While the overall content is impressive, I would like to bring to your attention two minor points for consideration:

  1. On lines 36-41, it would be beneficial to refer to a source to support the this statement.

  2. On line 385, it would be valuable if you could provide examples of other areas where your proposed approach could be implemented. Additionally, supporting these examples with relevant references would add depth to your discussion and provide readers with further avenues for exploration.

  3. If possible, consider incorporating references that offer a broader perspective on the methodologies discussed or that provide historical context for the issues addressed in your paper. Additionally, citing recent studies or advancements in the field could further demonstrate the relevance and timeliness of your research.

Thank you for your time and dedication to advancing research in this important area. I look forward to seeing the final version of your work.

Best regards,

Author Response

Dear Reviewer,

We would like to express our gratitude for your insightful comments and suggestions regarding our manuscript. Your feedback has been invaluable in strengthening the quality and depth of our paper. Below, we address each of your points and outline the changes made in response to your review.

  1. Reference for Statement on Lines 36-41
    • You suggested adding a source to support the statement in lines 36-41. We have now included a reference which provides empirical evidence supporting our claim. This addition not only validates our statement but also offers readers a direct resource for further reading on this topic.

  1. Examples and References for Broader Application of SCT
    • In response to your suggestion, we have expanded the discussion to include examples of SCT's application in other areas of healthcare education, such as nursing and pharmacy. We have also supported these examples with relevant references. This addition not only broadens the scope of our discussion but also provides readers with concrete examples and further avenues for exploration.

  1. Incorporating Broader Perspectives and Historical Context
    • To address your recommendation for including references that provide a broader perspective and historical context, we have added a new paragraph in the discussion section. This paragraph traces the evolution of clinical reasoning assessment from oral examinations to the current use of digital technology. We have cited key studies, to showcase both the historical development and recent advancements in the field. These citations not only enrich the historical context of our discussion but also underscore the relevance and timeliness of our research.

We are grateful for the opportunity to enhance our manuscript and appreciate your constructive feedback. The revisions made have undoubtedly improved the depth and clarity of our paper, and we believe these changes will make a valuable contribution to the field.

Thank you once again for your time and effort in reviewing our work. We look forward to the possibility of our revised manuscript being published in Healthcare journal.

Best regards,

Authors

Round 2

Reviewer 1 Report

Comments and Suggestions for Authors

I appreciate the efforts made by authors in addressing the issues raised, and I believe that they were solved. However, there is a major problem in the provided review: in the responses the direct reference to the new version of the manuscript (punctual indication of parts of the text removed, added or reworked) is not provided. As a consequence, it is very hard to assess if some of the comments were actually solved in the new version of the manuscript.

Dear Reviewer,

MAJOR COMMENTS

1) The main critical issue is related to the sample size: 55 tested subjects. While authors acknowledge the limitations of pooling the sample from a very characterized population (a single faculty in a specific university), and state that << the relationships observed in our study should not be interpreted as conclusive cause-and-effect phenomena but rather as indications of potential patterns that require further investigation>>, the limited sample size is not even cited in the limitations section. While the significance threshold is set by editors, I think that, at least, it necessary to elaborate this limit in the dedicated section, and to moderate and circumscribe the terms of the study’s validity. For example, statements such as:

·       Line 355: <<This study is a milestone in the journey to unravel this elusive process>>

·       Line 428: <<notable results>>

are inappropriate and should be moderated.

Sample Size and Study Validity Moderation

We have now included a more comprehensive discussion of the limitations related to our sample size in the limitations section. Additionally, we have moderated the terms used in the manuscript to reflect the preliminary nature of our findings. Statements such as “This study is a milestone...” and “notable results” have been revised to convey a more appropriate level of confidence.

Checked.

2) Study objects should be addressed more clearly. On one side, SCT is assessed in its capability to meaningfully evaluate student’s knowledge, while on the other side it is also studied how it can impact the learning path, possibly enhancing student’s propensity to learn. These two distinct aspects, both targeted in the study, should be separately stated more clearly, with their own specific research questions, results, and conclusions. In particular, I think that a more clear classification of the presented material (especially in the discussion section) towards one goal or the other would be strongly beneficial for the readability of the manuscript.

Study Objectives Clarity

We have revised the Discussion section to clearly differentiate the two main objectives of our study – evaluating SCT’s capability in assessing student knowledge and its impact on the learning path. Specific paragraphs have been added to address each aspect separately, with their own research questions, results, and conclusions.

Please indicate punctually the changes (parts of the text removed, parts of the text added) made to address this issue.

3) The link between ‘disaster preparedness’ and the implementation of SCT evaluation is not addressed with sufficient extension to justify the presence of this terminology in the manuscript title. It should be modified.

Manuscript Title Modification

Acknowledging the insufficient connection between ‘disaster preparedness’ and the implementation of SCT in our study, we have revised the title to “The Impact of Script Concordance Testing on Clinical Decision-Making in Paramedic Education” and made corresponding adjustments in the abstract.

Checked.

4) Figure 2: I suggest adding the excluded students (and the decisional criterion) at each step.

Figure 2 Revision

We acknowledge this suggestion and will add the excluded students and the decisional criteria at each step in Figure 2 in our subsequent revision.

Thank you for the rework. I suggest fixing the graphics, which is currently a little unclear.

5) The recorded differences between CI and CII should be addressed and elaborated in the discussion section.

Differences Between CI and CII: In the Discussion section, we have added a new paragraph specifically addressing and elaborating on the differences observed between Cohorts I and II, exploring potential reasons behind these differences.

Checked.

MINOR COMMENTS

6) Lines 80-85: this paragraph seems to be related (and partially superimposed) to that at lines 59-64. I suggest unifying the two.

Unification of Overlapping Paragraphs

The overlapping content in lines 80-85 and 59-64 has been unified into a single, coherent paragraph to eliminate redundancy.

Please indicate punctually the changes (parts of the text removed, parts of the text added) made to address this issue.

7) Lines 148-165: the first inclusion criterion states <<The first criterion was that all participants must be current undergraduate paramedic students at MUL, with no prior certification in qualified first aid>> and the first exclusion criterion is <<The first exclusion criterion stated that any participants who had previously received a diploma certified training in qualified first aid were excluded from the study.>> This seems somehow repetitive. The same goes for the second exclusion criterion.

Repetition in Inclusion and Exclusion Criteria

We have revised these sections to remove the repetitive content, streamlining the inclusion and exclusion criteria for clarity.

Please indicate punctually the changes (parts of the text removed, parts of the text added) made to address this issue.

8) Lines 207-209: <<The SCT completion was designed to be student-friendly, encouraging them to complete the test at their convenience in a comfortable environment. This approach was chosen to elicit accurate responses and minimize stress.>> This is an opposite condition with respect to the simulated situation. Why was this approach chosen? I think it is worth some elaboration.

Elaboration on SCT Completion Approach

We have elaborated on the rationale behind the student-friendly approach of SCT completion, explaining why this method was chosen despite its contrast with simulated paramedic situations.

Please indicate punctually the changes (parts of the text removed, parts of the text added) made to address this issue.

9) Lines 228-230: <<For quantitative variables, we used measures like mean value (M), standard deviation (SD), median (Me), minimum (Min), and maximum (Max) to describe central tendency and dispersion. The Shapiro-Wilk test was employed to assess the normality of data distribution, with a significance level set at p<0.05>> I understand that mean and standard deviation is used for normally distributed data, and median and interquartile range for non-normal data, as reported in line 261, is this correct? If so, I think it is necessary to add the IQR to this list.

Inclusion of IQR in Data Analysis

In line with your suggestion, we have included the Interquartile Range (IQR) in our data analysis section and adjusted the manuscript accordingly.

Please indicate punctually the changes (parts of the text removed, parts of the text added) made to address this issue.

10) Lines 277-283: thus paragraph is a repetition of 267-273.

Removal of Repetitive Paragraph

The repetitive content in lines 277-283 has been removed to improve the flow and coherence of the manuscript.

Please indicate punctually the changes (parts of the text removed, parts of the text added) made to address this issue.

11) Table 2: shouldn’t IQR data be presented instead of STD?

Revision of Table 2

In response to your suggestion, Table 2 has been revised to present IQR data instead of STD, providing a more appropriate measure of data variability.

Checked.

12) Lines 298-301: this part of the analysis should be anticipated in the methods section.

Introduction of Analysis Method in Methods Section

We have added a clear statement regarding the use of Cronbach’s alpha coefficient in the "Data Analysis" subsection of the Methods section, addressing the previously missing introduction of this part of the analysis.

Please indicate punctually the changes (parts of the text removed, parts of the text added) made to address this issue.

Comments on the Quality of English Language

Minor typos

Author Response

Dear Reviewer,

Thank you for your thorough review. Below, we detail how we have addressed each major and minor comment with text extracts.

MAJOR COMMENTS

  1. Study Objectives Clarity

We have revised the Discussion section to clearly differentiate the two main objectives of our study – evaluating SCT’s capability to assess student knowledge and its impact on the learning path. Specific paragraphs have been added to address each aspect separately, with their own research questions, results, and conclusions.

Clinical reasoning forms the backbone of healthcare, enabling professionals to make timely and beneficial decisions [19]. However, many factors influence clinical reasoning, making it a multifaceted, complex, and often elusive process [20]. Recognizing these challenges, our study aims to contribute to the ongoing exploration of clinical reasoning. Through a prospective cohort study involving a specific group of 55 paramedic students, we explored the utility of SCT in stimulating and evaluating clinical reasoning. While our findings provide valuable insights, they should be viewed as preliminary, given the limited sample size and the specific context of a single faculty at a particular university. As such, this study represents an exploratory step in understanding the application of SCT in paramedic education.

  1. Figure 2 Revision

We acknowledge this suggestion and will add the excluded students and the decisional criteria at each step in Figure 2 in our subsequent revision.

We have cleaned up Figure 2.

  1. Differences Between CI and CII: In the Discussion section, we have added a new paragraph specifically addressing and elaborating on the differences observed between Cohorts I and II, exploring potential reasons behind these differences.

In our analysis, notable differences were observed between Cohort I (first-year students) and Cohort II (second-year students) in their response to and performance on the SCT. These variations may be attributed to several factors, including the differing levels of exposure and experience with clinical environment and decision-making processes. Second-year students, having had more time to acclimatize to the academic and practical aspects of paramedic training, might be better equipped to handle the complexities of the SCT. This disparity underscores the importance of progressive and scaffolded learning approaches in paramedic education, where students gradually build their competencies over time. Additionally, these findings suggest the need for early and continuous exposure to diverse assessment tools, like the SCT, throughout the paramedic curriculum. This approach could facilitate a more uniform development of clinical reasoning skills, bridging the gap observed between different year cohorts.

MINOR COMMENTS

  1. Unification of Overlapping Paragraphs

The overlapping content in lines 80-85 and 59-64 has been unified into a single, coherent paragraph to eliminate redundancy.

Those two paragraphs were merged and rephased as one:

Furthermore, the impact of the SCT on students’ learning experiences, particularly in paramedic education, requires further exploration. Studies suggest that the SCT positively influences the learning process, aiding in self-evaluation and identifying knowledge gaps. However, its acceptability among students needs more understanding [5,6]. Moreover, assessing students' attitudes towards SCT is crucial. If they do not perceive it as beneficial, its effectiveness might be compromised [7-9]. Exploring these attitudes can offer insights into SCT's optimal integration into curricula and enhancing learning experiences. This exploration is particularly vital given the unique challenges and high-stress nature of paramedic work, where decision-making skills are paramount.

  1. Repetition in Inclusion and Exclusion Criteria

We have revised these sections to remove the repetitive content, streamlining the inclusion and exclusion criteria for clarity.

2.3.1. Inclusion Criteria

The study focused on undergraduate paramedic students at MUL who were in their first or second year and were either due to undertake or had already completed the qualified first aid course. This ensured that participants were at a similar level of initial knowledge, making the results more comparable. Additionally, all participants were required to have completed an introductory first aid training course prior to the qualified first aid course, ensuring a foundational understanding of first aid principles.

2.3.2. Exclusion Criteria

Participants were excluded if they had previously received diploma-certified training in qualified first aid, to avoid the confounding effects of advanced prior knowledge. Additionally, individuals with other certifications related to paramedic qualifications, such as a college paramedic diploma, were excluded. This was to ensure that additional qualifications did not influence their performance in the SCT or their progression during the first aid course.

  1. Elaboration on SCT Completion Approach

We have elaborated on the rationale behind the student-friendly approach of SCT completion, explaining why this method was chosen despite its contrast with simulated paramedic situations.

The SCT completion process was intentionally designed to be student-friendly. We chose to allow students to complete the test at their convenience in a comfortable environment, diverging from the high-pressure, unpredictable scenarios typically associated with paramedic work. This approach was adopted to ensure the accuracy of responses and minimize stress, which can significantly influence cognitive performance. By providing a relaxed setting, we aimed to get a clear measure of their clinical reasoning skills without the confounding effects of stress or time pressure.

  1. Inclusion of IQR in Data Analysis

In line with your suggestion, we have included the Interquartile Range (IQR) in our data analysis section and adjusted the manuscript accordingly.

For quantitative variables, we used measures such as mean value (M), standard deviation (SD), median (Me), interquartile range (IQR), minimum (Min), and maximum (Max) to describe central tendency and dispersion. The inclusion of the IQR was particularly crucial for data not following a normal distribution.

  1. Removal of Repetitive Paragraph

The repetitive content in lines 277-283 has been removed to improve the flow and coherence of the manuscript.

Removed

To investigate the concurrent validity further, we conducted Bland-Altman plots comparing SCT and MCQ examination results. The analysis indicated a mean difference of about 14% between the results of the two methods, with MCQ results being, on average, 14% higher than those of the SCT. However, the wide limits of agreement (LOA ± 1.96 SD: 41.7% to -13.4%), and the fact that zero is within these limits, suggest there is no meaningful difference between the two measures. The LOA was also visibly dispersed, indicating a weak concurrent agreement between the SCT and MCQ scores.

Included:

To investigate the concurrent validity further, we conducted Bland-Altman plots comparing SCT and MCQ examination results (Figure 2). The analysis indicated a mean difference of about 14% between the results of the two methods, with MCQ results being, on average, 14% higher than those of the SCT. However, the wide limits of agreement (LOA ± 1.96 SD: 41.7% to -13.4%), and the fact that zero is within these limits, suggest there is no meaningful difference between the two measures. The LOA was also visibly dispersed, indicating a weak concurrent agreement between the SCT and MCQ scores (Table 1).

  1. Introduction of Analysis Method in Methods Section

We have added a clear statement regarding the use of Cronbach’s alpha coefficient in the "Data Analysis" subsection of the Methods section, addressing the previously missing introduction of this part of the analysis.

Importantly, for evaluating the internal consistency of the SCT, we used Cronbach’s alpha coefficient. This metric was critical to assess the reliability and homogeneity of the test items within the SCT, ensuring the assessment's robustness and appropriateness for our study objectives.

We believe that we have addressed the concerns you have raised. We are grateful for your feedback and hope that our manuscript is now suitable for publication.

Thank you for your consideration.

Sincerely,

Authors
